# Work-Life Conflict among U.S. Long-Haul Truck Drivers: Influences of Work Organization, Perceived Job Stress, Sleep, and Organizational Support

**DOI:** 10.3390/ijerph16060984

**Published:** 2019-03-19

**Authors:** Adam Hege, Michael K. Lemke, Yorghos Apostolopoulos, Brian Whitaker, Sevil Sönmez

**Affiliations:** 1Public Health Program, Department of Health & Exercise Science, Appalachian State University, Leon Levine Hall, 1179 State Farm Road, P.O. Box 32071, Boone, NC 28607, USA; 2Department of Social Sciences, University of Houston-Downtown, One Main Street, Houston, TX 77002, USA; lemkem@uhd.edu; 3Complexity & Computational Population Health Group, Texas A&M University, College Station, TX 77843, USA; yaposto@hlkn.tamu.edu; 4Department of Health & Kinesiology, Texas A&M University, College Station, TX 77843, USA; 5Department of Management, Appalachian State University, 416 Howard Street, P.O. Box 32089, Boone, NC 28608, USA; whitakerbg@appstate.edu; 6College of Business Administration, University of Central Florida, 12744 Pegasus Drive, Orlando, FL 32816, USA; sevil@ucf.edu

**Keywords:** long-haul truck drivers, work-life balance, work organization, sleep, job stress, occupational health disparities

## Abstract

Work-life balance and job stress are critical to health and well-being. Long-haul truck driving (LHTD) is among the unhealthiest and most unsafe occupations in the U.S. Despite these disparities, there are no extant published studies examining the influence of work, stress and sleep outcomes on drivers’ work-life balance. The current study investigated whether adverse work organization, stress, and poor sleep health among LHTDs are significantly associated with work-life conflict. Logistic regression was used to examine how work organization characteristics, job stress, and sleep influenced perceived stress and a composite measure of work-life conflict among a sample of 260 U.S. LHTDs. The pattern of regression results dictated subsequent analyses using structural equation modeling (SEM). Perceived job stress was the only statistically significant predictor for work-life balance. Fast pace of work, sleep duration and sleep quality were predictors of perceived job stress. SEM further elucidated that stress mediates the influences of fast work pace, supervisor/coworker support, and low sleep duration on each of the individual work-life balance indicators. There is an urgent need to address work conditions of LHTDs to better support their health, well-being, and work-life balance. Specifically, the findings from this study illustrate that scheduling practices and sleep outcomes could alleviate job stress and need to be addressed to more effectively support work-life balance. Future research and interventions should focus on policy and systems-level change.

## 1. Introduction

The last four decades have been marked by drastic changes to work and employment conditions in the U.S. and globally [1]. In turn, American workers are working longer hours, encountering upsurges in shift work experiences, facing increasing burdens of psychosocial job stressors, and suffering significant work-life imbalances [2,3,4]. Considering the poorer health outcomes in the U.S. compared with most other developed nations, it is becoming increasingly urgent to examine work as a major social determinant of health [1,5,6,7].

Work organization, shaped by a combination of macro-, meso- and micro-level forces, has been shown to have profound health impacts and to serve as a significant contributor to occupational health disparities [8,9]. At the behavioral level, adverse work environments have been associated with risky health behaviors [10,11], while also influencing outcomes such as obesity and cardiometabolic disease [12,13,14,15,16], sleep [17,18], and mental illness [19,20]. Due to numerous psychosocial and physical risk factors, studies have shown that the occupational sectors most at risk for health disparities include: transportation; agriculture; construction; and healthcare [8].

There are nearly two million U.S. long-haul truck drivers (LHTDs), most of whom are middle-aged, White, and married, although they endure marital and family strain due to their job demands [21,22,23]. Long-haul truck drivers spend long periods of time away from home, traversing American interstates daily with work conditions, such as scheduling, which are largely out of their immediate control. In fact, the trucking industry makes up the largest segment of the transportation sector, while the work of a LHTD has been described as a “sweatshop on wheels” [24]. Linked closely with the industry’s work organization, work stressors have been associated with numerous poor health outcomes and highway accident risks, which have considerable public health and societal implications [22,23,25,26,27].

While research related specifically to the work of U.S. LHTDs is limited, some researchers have explored connections between work-life balance, or what is often referred to as work-family or work-life conflicts, and health and quality of life outcomes in other occupational contexts [28,29,30,31,32,33,34,35]. In general, work-life balance, which encompasses both work-family and work-life conflicts, is a term used to describe the balance that individuals need between the time allocated for work and other aspects of life, including family, social and leisure pursuits, and other domains of health and well-being [35]. Not surprisingly, employees with work organizations requiring long work hours, minimal time off, and other poor work conditions are more likely to report work-life imbalances or work-life conflict [36,37]. Furthermore, workers with a work-life conflict also tend to exhibit negative health behaviors [38,39] and outcomes such as insufficient sleep [40,41,42] and mental illness (e.g., anxiety, depression) [34,43,44].

Recent media coverage of the commercial trucking sector has drawn attention to the fact that many LHTDs are unwilling to join or remain in the profession due to poor working conditions and that the future of transporting goods across the nation could be in dire need for change—much of which is related to the chronic work-life conflict and the health and safety risks that come with the profession [45,46]. A great deal of research attention on LHTDs and other American workers has focused on poor sleep outcomes in relation to work organization and job stress [47,48,49,50]. There is less understanding of the connections between work organization, sleep outcomes, perceived job stress, and work-life balance. It is plausible that work stressors, or perceived job stress, serves as a mediator between work and sleep, thereby having substantial impact on life outside of work and furthermore, sleep could have a direct impact on perceptions of work-life conflicts [49,50,51].

Undeniably, work in the new 24/7 economy has significant population health consequences in the U.S. and the work of LHTDs presents a unique but vital occupational context. While there has been an increase in research related to health behaviors and outcomes of the LHTD population in connection with the work conditions, we are aware of no previous research that has been specifically focused on the impact of work-life conflicts in the population. While not in the LHTD population, Williamson and colleagues [52] reported that short-haul drivers in Australia who reported an excess of work-life conflict were much more likely to also experience work-related injuries and illness. It is plausible, however, also to reason that stress and poor sleep associated with work conditions would influence how drivers perceive their ability to have an adequate work-life balance.

Extant theoretical frameworks regarding work-life conflict have not been used to explore these connections in the context of LHTD. In their seminal review paper, Puttonen and colleagues [53] posited that the combination of excess work demands and work stress are associated with poorer sleep (both duration and quality), and this is most likely due to a lack of work-life balance or “recovery” period. The researchers further reported that poor work schedules (long work hours, shift work), specifically, lead to work stress, work-life conflicts, and poorer sleep. With LHTDs working long hours, always rushing to meet work demands and working in a stressful environment, and having irregular work and sleep schedules during the day and night, it is expected that when they are at home, drivers will be catching up on missed sleep and rest to recover and prepare for their next trip. This will interfere with many of their out-of-work activities—in effect, one would expect that this combination of work demand, stress, and poor sleep would play a large role in predicting how drivers perceive their ability to have a work-life balance. In turn, as Puttonen [53] hypothesized, the poor sleep outcomes could potentially exacerbate how drivers perceive their overall job stress and their work demands.

In addition to the aforementioned dearth of studies exploring work-life conflict theoretical frameworks in the context of LHTD, existing research suggests that existing theory may be insufficient to capture these complex relationships in this unique occupational milieu. Investigations into work-life conflict among nurses, who share several detrimental work organization challenges (especially frequent shift work and long work hours) with LHTD, have suggested that current theories do not fully explain the relationships among work-family conflict factors and sleep outcomes [42]. Furthermore, other studies have highlighted the complex and often bewildering connections between work-family conflict and sleep outcomes. For example, among information technology workers, work-to-family conflict, family-to-work conflict, and family supportive supervisor behaviors were associated with sleep duration and sleep quality, although several of these connections were surprising, with work-to-family conflict negatively associated with sleep duration while family-to-work conflict was not [54].

One such theoretical framework that helps to explain the relationship between occupational stresses, the adverse effects on sleep, and subsequently work-life conflicts is the Conservation of Resources (COR) theory [55,56,57]. COR places emphasis on the role that human behavior is largely predicated on our ability to attain and maintain resources; specifically, resources can be both internal (i.e., hope, self-efficacy) or external (i.e., employment conditions, social support, family, health) [56]. When it comes to the issue of sleep among LHTDs, it becomes a valuable resource for them in terms of their ability to perform their job and have a quality of life outside of work; however, the long hours of work and stress placed on them on a daily basis makes adequate sleep much hard to attain [57]. In effect, with most drivers paid by the mile, LHTDs are incentivized to work longer and drive further to increase their income; this often comes at the expense of sleep. Therefore, drivers tend to have to “catch up” on sleep on their non-working days, which affects their ability to engage with other valuable resources (i.e., family, health, social/leisure activities).

Previous LHTD research [58,59] has used mediation and moderation modeling to explore the influence of the organizational and policy climate, including supervisor and organizational support (for LHTDs it is typically provided primarily by the scheduling dispatcher), on how drivers perform relative to safety. LHTDs are considered ‘lone workers’, in that most of their work duties are performed without the typical support provided by interaction with co-workers. From a theoretical perspective, using previous literature from the fields of organizational psychology and occupational safety and health, Zohar and colleagues [58,59] tested and found that how drivers perceive their supervisor determines how they also view the safety climate; in addition, these mediate how drivers perform on an individual level when it comes to safety practices. This could further be adapted and examined in relation to how drivers perceive their job stress, sleep outcomes, and work-life conflict.

With this context of the occupational milieu of LHTDS and grounded in the COR theoretical framework, this study sought to explore relationships between work, sleep, perceived stress, and their subsequent impacts on the work-life conflict of a sample of LHTDs. Specifically, researchers were interested in exploring the influence of work stressors and sleep challenges on how drivers perceive their work-life conflicts and job stress. The current study had two primary hypotheses that were tested using logistic regression:
(1)that the combination of adverse work organization characteristics and an increased perceived job stress serve as a predictor of an increase in driver’ reporting of sleep negatively impacting their work-life balance;(2)and that the combination of adverse work organization characteristics and poorer sleep outcomes are predictors of higher perceived job stress.

In addition, with the literature supporting the notion that organizational support and safety climate could impact the aforementioned relationships, and in a post-hoc response to the regression findings, we used structural equation modeling (SEM) to test a path analysis.
(3)we hypothesized that occupational stress mediates the relationship between on-the-job factors (scheduling, supervisor support), sleep, and subsequent work-life conflict.

A visual representation (Figure 1) of our hypotheses is found below.

## 2. Materials and Methods

### 2.1. Study Setting and Sample

Data collection for this study of U.S. LHTDs took place at a large truck stop located in central North Carolina. The methods have been further detailed in previous papers that resulted from this data set [60,61]. In brief, the study used a cross-sectional, nonexperimental study design and an interviewer-administered Truck Sleep Disorders Survey (TSLDS) to collect data from 262 drivers. Prior to the study, researchers performed power analyses to determine the appropriate sample size. The survey instrument was developed from other key instruments related to work, sleep, health, and our previous work with truck drivers [62,63]. Due to missing data from two of the drivers, a final sample size of 260 drivers was achieved for statistical interpretation. The study was approved by the Institutional Review Board (IRB) at the University of North Carolina at Greensboro (12-0248).

### 2.2. Study Measures

**Work organization.** Drivers were asked a series of questions about characteristics related to their work. Key features included: the number of days drivers were on the road per month, the number of daily work hours, irregularities within daily and weekly work schedules, pace of work and experiences with time pressures, and support systems such as coworkers and supervisors. For days on the road, drivers were asked about a five-day sequence with possible answers ranging from *less than 5 days, 6–10 days, 11–15 days, 16–20 days, 21–25 days, 26–30 days, over one month,* to *more than two months*. With so few drivers reporting 20 or fewer days, researchers grouped the variable for analysis as *0 = 20 or less days, 1 = 21–25 days,* and *2 = 26 or more days*. Likewise, with work hours, drivers were asked about a one-hour sequence. With so few drivers having a lower number of work hours, this variable was grouped as *0 = less than 11 h, 1 = 11–13 h*, and *2 = 13 or more hours*. Drivers were asked about specific experiences with shift work indicators, including the irregularity of their daily and weekly schedules. Possible responses included *same each day/week* or *different each day/week.* Regarding work pressures and social support systems, drivers were asked about their frequency of fast pace and support of coworkers and supervisors, with response selections of *never or rarely, sometimes,* or *often or always.*

**Stress and sleep outcomes.** For perceived stress, response selections included *none, mild, moderate, high,* and *extreme or chronic.* Based on the breakdown of the data, these were grouped as *none-mild, moderate,* and *high-chronic*. Sleep duration, on both workday and non-workdays, was asked in terms of the number of hours of sleep for an entire 24-h period. Specifically, drivers were asked to characterize their sleep over the past two weeks. Drivers were also asked about the number of hours they felt they needed to achieve the highest function possible, in order to determine possible gaps between the amount of sleep achieved and what was desired. Lastly, sleep quality, on both workdays and non-workdays, was determined with the question *How often do you feel that you get a good night’s sleep?* with possible answers of *never or rarely* and *almost every night or every night*.

**Impact of sleep on drivers’ work-life conflict.** Drivers were asked about the effects of their sleep on their jobs, as well as on aspects of their lives outside of the workplace. In a series of seven questions with the same responses (*no impact, some impact, major impact*), drivers were asked about the impact of their sleep on their work, their social and leisure activities, family and home responsibilities, mood, intimate and sexual relationships, physical health, and mental health. Based on the literature [34,35], these questions were considered indicators of impact on work-life conflict. The reliability of the scale of questions was robust (Chronbach’s alpha = 0.90), indicating a strong association. Based on this, a composite variable (work-life conflict) was created for further analyses, with a possible score ranging from zero to 14 (0–2 for each of the seven variables).

### 2.3. Statistical Analysis

Descriptive analyses of all of the variables was completed at the start, to give a broad overview of study respondents’ characteristics (Table 1, Table 2 and Table 3). With the aforementioned work-life conflict composite variable, frequencies were grouped into quartiles, which became *none to minor impact* (0–3), *mild impact* (4–6), *high impact* (7–9), and *major impact* (10 or greater). To test our two primary hypotheses, relationships between work organization characteristics, sleep duration and quality, stress, and the aforementioned composite work-life conflict variable were examined via two ordinal logistic regression analyses, while controlling for age and length of tenure as a driver. The first model (Table 4) featured work-life conflict as the outcome variable and the following predictors: days on the road per month, daily work hours, regularity of daily and weekly schedules, frequency of a fast work pace, support provided by supervisor/dispatcher, and perceived stress. Sleep duration and sleep quality were removed from the group of predictor variables because the impact of work characteristics on the interaction between sleep and work-life conflict was the primary concern. Sleep was already accounted for in the work-life composite variable. In addition, we tested the model with sleep duration and sleep quality variables prior to finalizing the model, and it was not statistically significant (X = 20.68, *p* = 0.15); however, when removing the sleep variables, the model was statistically significant (X = 34.39, *p* < 0.001). In the second model (Table 5), perceived stress served as the outcome variable and with the same predictors as in the first model, with the addition of sleep duration and sleep quality variables. Based on our understanding of the profession and findings of previous studies, we did not believe coworker support warranted inclusion as a potential predictor, primarily because LHTDs spend most of their time alone and have little interaction with coworkers. Dispatchers, who determine and communicate driving schedules to drivers, also serve as their supervisors and are primarily who LHTDs communicate with while on the road. All descriptive statistical analyses were performed using SPSS 23.0 (Armonk, NY, USA) [64].

From our logistic regression findings, we were interested in exploring potential mediating relationships between work organization characteristics, sleep outcomes, stress, and drivers’ perceptions of their sleep’s impact on work-life conflict. Although structural equation modeling (SEM) is typically conducted with longitudinal data and SEM with cross-sectional data cannot establish causal pathways, previous studies have effectively used SEM in post-hoc analyses involving cross-sectional data used for testing the directional associations between variables and the fit of a hypothesized model [65,66,67,68,69,70]. Based on the regression analyses, SEM was used to explore for potential mediating influences (Figure 2). We made use of Mplus for SEM purposes [71].

With our SEM analyses, we included the combination of supervisor and coworker support as a factor that could affect these relationships through the concept of work social support; in their meta-analysis, Michel and colleagues [72] found that this social support system in the workplace can predict work-life balance outcomes or a “work-to-family conflict”. Furthermore, Kossek’s [73] meta-analysis concluded that supervisor support is more related to how employees perceive their work-life conflict than the comprehensive support system (i.e., scheduling practices) provided by the employer, thereby indicating that supervisor support and a positive relationship is vital to work-life balance. This is especially important in the LHTD profession due to the fact that drivers spend most of their time isolated and alone, with much of their human interaction occurring with their dispatcher, who serves as their immediate supervisor. Supportive supervisors are also more likely to strive to make sure drivers have a schedule that promotes their health and wellbeing. In addition, drivers have very little interaction with coworkers, outside of the truck stop setting. Therefore, we hypothesized that the support received from their supervisors, and limited support from coworkers (that is often found in other professions), can play an impact on how drivers are impacted and how they perceive the stress, poor sleep outcomes, and sleep’s influence on work-life conflicts.

## 3. Results

Descriptive statistics regarding work organization are provided in Table 1. More than four out of every five drivers (84.6%) reported spending 21 or more nights away from home each month and 70.4% reported working 11 or more hours daily. Greater irregularity was found in the daily scheduling of drivers when compared to the weekly irregularity. More specifically, 82.7% of the study participants reported working a different daily schedule each day, 63.8% reported working an irregular number of hours daily, while only 32.4% reported a varying weekly schedule. Drivers experienced a relatively high frequency of fast pace of work (68.0% doing so at least some of the time). Lastly, drivers reported high levels of support from their supervisors (76.2% had support often or always) and moderate levels of support from their coworkers (48.9% had support often or always). 

Drivers’ stress and sleep outcomes, as well as their impacts, are presented in Table 2 and Table 3. In terms of stress, 62.6% of drivers felt their stress level was moderate or high. A wide discrepancy was found in drivers’ sleep duration on workdays (6.95 h) compared to non-workdays (8.27 h). Linked with sleep duration, drivers reported much better sleep quality occurring on their non-workdays when compared to workdays. More specifically, 38.2% reported never or rarely getting a good night’s sleep on workdays, whereas only 16.7% did so on their non-workdays. 

Sleep had significant impacts on drivers’ work performance as well as their non-work-related activities. Specifically, 71.1% felt sleep had at least some impact on their work. In terms of the impact outside of work, 58.6% reported at least some impact on social and leisure activities, 59.1% reported at least some impact on family and home responsibilities, 82.0% reported at least some impact on their mood, 48.1% reported at least some impact on their intimate and sexual relationships, 63.0% reported at least some impact on their physical health, and 62.2% reported at least some impact on their mental health. When examining the aforementioned work-life composite variable, the mean score was 6.43, ranging from zero to 14. Researchers examined the quartile breakdown of the sample and nearly half of all drivers (48.6%) were impacted by sleep at a high or major impact level.

Logistic regression results with the work-life composite variable as the outcome are presented in Table 4. The model was statistically significant (X^2^ = 34.39, *p* < 0.001). The only statistically significant predictor for a worse work-life conflict due to sleep was perceived stress, with mild stress or less and moderate stress both holding a 55% reduction in odds when compared to high or chronic stress. While not statistically significant, never or rarely experiencing a fast pace of work again led to a reduced impact (OR = 0.38), suggesting that the pace of work could be impacting this relationship between perceived stress and work-life conflict.

Logistic regression results with perceived stress as the outcome are presented in Table 5. The model is statistically significant (X^2^ = 65.01, *p* < 0.001) with significant predictors to the model including the frequency of fast pace of work, sleep duration on both workdays and non-workdays, and sleep quality on non-workdays. Specifically, never or rarely having a fast pace of work held 82% reduced odds, and sometimes having a fast pace of work held 76% reduced odds when compared to often or always having a fast pace of work for an increased level of stress. Sleep duration led to some interesting results. Increased sleep duration on workdays led to a 40% reduction in odds of an increased stress level; whereas, increased sleep duration on non-workdays (OR = 1.27) led to increased odds of higher stress. Lastly, good sleep quality on non-workdays predicted a 68% reduction in odds for increased stress levels. The combination of increased sleep duration and improved sleep quality on non-workdays implies that “catching up” on sleep on non-workdays is critical to drivers; however, it can interfere with their non-work-related responsibilities and leisure activities as a result. 

To simultaneously examine relationships between work conditions, sleep, perceptions of support, stress, and work-life conflict indicators, as well as to generate standardized coefficients describing these relationships, a SEM was tested to investigate job stress status as a critical mediating mechanism (Figure 2). Mplus v 4.21 [74] was used to fit the just-identified model to the covariance matrix, which resulted in acceptable fit as evidenced by fit indices; χ^2^(1) = 2.36, *p* = *ns*; CFI = 0.93; RMSEA = 0.09, SRMR = 0.01. To create the most parsimonious model, two independent variables were specified in order to represent work conditions (fast pace of work; supervisor/coworker support) and one to represent sleep issues. Supervisor/coworker support was a composite variable created by the two questions on how often respondents feel supported by their supervisors and coworkers. Sleep gap was operationalized as the numerical difference between the number of hours of sleep needed for highest function and number of hours of sleep respondents actually achieve on work nights. This particular measure has been utilized by the National Sleep Foundation when analyzing sleep duration (and quality) in relation to health outcomes in their Sleep in America polls [75]. 

Standardized paths from the SEM are presented in Figure 2. As shown, faster pace of work was significantly and positively associated with stress status (*b* = 0.26, *p* < 0.05), while larger sleep gaps and supervisor/coworker support were significantly and negatively related to stress status (*b* = −0.31, *p* < 0.05 and *b* = −0.18, *p* < 0.05, respectively). These results indicate that stress status, at least partially, mediates the influence of fast pace of work, sleep gaps, and supervisor/coworker support on each of the work-life conflict indicators. 

## 4. Discussion

The unique occupational milieu of the U.S. long-haul trucking industry, and especially its work organization and workplace characteristics, has been known to have detrimental impacts on the physical and psychological well-being of its drivers [23,76,77,78,79,80,81]. In turn, these physical and psychological consequences—including poor work-life balance—are believed to reduce the economic vitality and viability of the industry, with hypothesized impacts on numerous stakeholders, including drivers and their families, trucking companies, health care systems, and other roadway users [23,82,83,84,85,86,87]. 

The LHTDs in our sample reported numerous detrimental impacts on their work-life balance due to poor sleep. Sleep sufficiency—and especially sleep quality—are notoriously poor in the U.S. long-haul trucking industry, even compared to other transport operator professions [88]. LHTDs typically get their rest in the sleeper berths of their truck cabs, with these sleep cycles being frequently interrupted by environmental factors (heat, cold, noise), dispatchers, and other factors [89]. In addition, various work organization (e.g., long work hours) and behavioral (e.g., low levels of physical activity) factors contribute to poor sleep health outcomes [90,91,92,93,94,95,96]. In our sample, poor sleep was especially attributed to specific elements of work-life balance among LHTDs, and in particular their overall work performance, mood, mental health, and physical health. It is notable that these work-life conflict outcomes are apparently those that are more proximal to workdays, while other work-life outcomes that are more distal from workdays, such as *impact on intimate and sexual relationships* and *impact on family and home responsibilities*, were less impacted by sleep. It is unclear why this difference was borne out in our analyses; it may be an artifact of how work-life stress was queried during data collection, or it may represent a true disparity in how work-life balance is perceived while drivers are working versus when they are at home.

### 4.1. Connections between Work Organization, Sleep, Stress and Work-Life Balance

As evidenced by our findings, it appears that the Conservation of Resources theory can be successfully applied to investigating relationships between work organization, stress, sleep, and work-life conflict. Our hypotheses were generally supported, although these connections and their apparent interrelationships were not as expected in several important ways. In the first regression model, which explored predictive relationships between work organization, stress, and work-life balance, perceived stress was the only statistically significant predictor. The strong relationship between perceived stress and work-life conflict was expected and has been found among LHTDs and other populations, and especially in terms of mental health outcomes such as depression, anxiety, and substance addiction [96,97,98,99,100,101,102,103]. The lack of additional statistically significant predictors related to work organization was surprising, and this supported our decision to explore whether or not stress functions as a mediating factor between work organization and sleep with work-life conflict.

Next, regarding perceived stress, frequency of fast pace of work emerged as a statistically significant predictor. This was expected, as similar relationships have been found among LHTDs other occupational segments such as package drivers, managerial and hourly hotel workers, and the catering industry [96,97,98,99,100,101,102,103,104,105]. These pace-of-work pressures may also inhibit coping behaviors, such as engaging in physical activity and achieving adequate sleep [106,107,108]. Although this was not explored in this study, the degree of perceived job control has been found in other studies to influence this relationship [109,110] and may play an important role in how LHTDs perceive their pace-of-work. In the long-haul trucking industry, important changes have occurred over the last three decades, such as the shift toward just-in-time deliveries, which have removed the degree of job control that LHTDs have in their work schedules [111,112]. Furthermore, compensation has shifted toward mile-based pay, with many driver job duties not typically compensated; as a result, LHTDs must work more hours than in the past to achieve desired levels of income [113]. 

Sleep duration and quality emerged as statistically significant predictors of perceived stress. The connections between sleep duration and stress are well-established in the scientific literature and many of the underlying physiological and psychological mechanisms have been identified [114]. Similarly, the connections between sleep quality and perceived stress have also been established, including among LHTD populations [115,116]. The connections between sleep duration, sleep quality, and stress, however, are somewhat complex among LHTDs. For example, while sleep sufficiency is known to improve the ability of drivers to cope with stressors, many drivers use other coping techniques, such as consuming caffeine or eating unhealthy “comfort food”, which compromise sleep health [117,118,119,120]. Additionally, sleep duration has itself been found to predict sleep quality [92]. Interestingly, a counterintuitive outcome was found in the present study: workday sleep duration was a protective factor against perceived stress, which was expected; however, non-workday sleep duration actually led to increased odds of higher perceived stress. One potential explanation may be rooted in the sleep duration question in the survey itself, which relied on self-reported hours of sleep on workdays and non-workdays. It may be that those drivers with poorer workday sleep durations were more likely to report higher levels of sleep duration on non-workdays, which would represent a cognitive bias rather than an actual negative impact of non-workday sleep duration. 

The SEM path model illuminates the complex connections between sleep, work organization, and work-life conflict, and confirms the potential role of perceived stress in mediating these connections. Our model’s results further support our hypothesis that the combination of insufficient and poor quality of sleep and high stress from the workplace are associated with significant impacts on health and life outside of work, resulting in an inadequate work-life balance. Here, pace of work and sleep duration were again shown to be significant factors in perceived stress, which reaffirms our findings from the regression model; however, supervisor/co-worker support was found to be a statistically significant factor in perceived stress, which was not the case in the prior regression model. This finding corresponds with prior studies that focused on LHTDs and other occupations [98,101,102,103,105]. It is possible that supervisor/co-worker support is especially important for LHTDs, whose work characteristics demand that they are away from their families and friends for weeks or even months at a time and experience excessive social isolation [86,98,99]. For LHTDs, their primary social interactions at work are with their dispatchers/supervisors and fellow drivers, which escalates the importance of these interactions in their overall social environment. Ultimately, this model highlights the inherent complexity in LHTD physical and psychological wellbeing, suggesting numerous interacting pathways across multiple levels of influence. This inherent complexity warrants further exploration, and these findings lend credence to calls from other authors to utilize complex systems approaches to map these and other connections and identify preventive solutions [113,114,115].

### 4.2. Recommendations to Improve Work-Life Balance

Considering the findings of the current study, which emphasize the key role of perceived stress as a mediator between the work organization and sleep of LHTDs and subsequent work-life balance, preventive interventions should be developed and proliferated that specifically target stress reduction. In addition, it is widely recognized that LHTDs lack many support and healthcare resources while out on the road away from home for extended periods of time. Characterized as ‘lone workers’, LHTDs often do not have the daily interactions with coworkers that can help to alleviate or buffer stress stemming from the workplace and strategies should be considered that can assist in addressing these elements of their work. These interventions should include components that overlap with other key LHTD health needs. For example, interventions that seek to increase levels of physical activity would simultaneously reduce stress, strengthen the body’s ability to deal with stressors, improve specific elements of work-life balance (e.g., reduced depression, improved mood), support sleep quality and duration, reduce fatigue and improve roadway safety, and enable better weight management and cardiometabolic disease prevention [116,117,118]. Similarly, interventions that target improved nutrition, better sleep health, sleep disorder screening and treatment, and substance prevention can have similar widespread impacts [120]. These suggested intervention components have been featured in existing LHTD workplace health and wellness programs [116,117,120] and represent prime initial pathways to improve work-life balance outcomes. At the same time, interventions should incorporate specific stress reduction components, such as teaching stress management techniques, educating drivers about health behavioral responses to stress, and providing stress-related mental health resources such as counselors [120]. However, to ultimately have long-term population level impacts, these interventions must be (a) comprehensive, unlike the bulk of LHTD intervention efforts that are usually siloed and limited in scope; (b) upstream in order to target those work organization forces (e.g., pace-of-work) that induce stress responses across the entire industry; and (c) accompanied by changes to LHTD worksite environments, which are notoriously unhealthy [119,120].

### 4.3. Limitations

There were three primary limitations of this study. First, the overall sample size is relatively small, with 262 long-haul truck drivers who participated in the survey, 260 of whom were included in these analyses. A larger sample size may have increased our likelihood of finding additional relationships, although similarities in our findings with other comparable studies that investigated LHTD indicates that our sample was highly representative of the population. Second, our instrument for measuring work-life conflict has not been validated, which indicates that caution must be taken in evaluating study findings, and our use of self-reported sleep measures is a limitation. Finally, selection bias may have occurred when recruiting drivers to participate in the survey. Our experiences with this population suggest that ambivalence or mistrust may exist in how LHTD perceive institutions such as universities, and as a result, may have led to systematic differences between drivers who participated and drivers who did not. 

## 5. Conclusions

Our findings corroborate a growing body of literature that indicates that work conditions have significant implications for health disparities in the U.S. and globally. Stress stemming from the workplace can significantly impact life outside of the workplace. An appropriate work-life balance and limited work-life conflict—largely nonexistent in the LHTD profession—is vital to improving the health and quality of life of all working populations. The effects of inadequate sleep, occupational stress and work-life conflicts applies to numerous other diverse occupations, including the healthcare industry, police officers, information technology, to name a few, particularly with new and emerging technologies that allow workers to work remotely and at all hours of the day or what is now referred to as ‘boundaryless’ work [31,41,121,122,123]. As described in the current study, the occupational milieu experienced by members of the long-haul trucking profession serves as a critical direct influence on the health and wellbeing of a vastly important population to the U.S. economy. This is not surprising, given the mounting evidence of how the occupation adversely affects one’s health and way of life. With the notion that work plays such an essential role in American life, it is imperative that policymakers address how to more effectively protect and promote the health and wellbeing of millions of vulnerable working populations, such as LHTDs. This research can and should be translated to other working populations as public health scientists continue to further understand the relationships between work and health. In addition, researchers must engage in advocacy efforts and continue to raise awareness about the connections between work and health.

## Figures and Tables

**Figure 1 ijerph-16-00984-f001:**
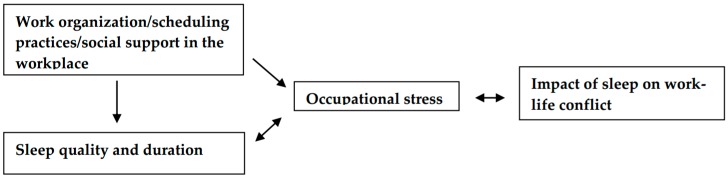
Hypotheses for relationships between work organization/scheduling, sleep, stress, and work-life conflict among LHTDs.

**Figure 2 ijerph-16-00984-f002:**
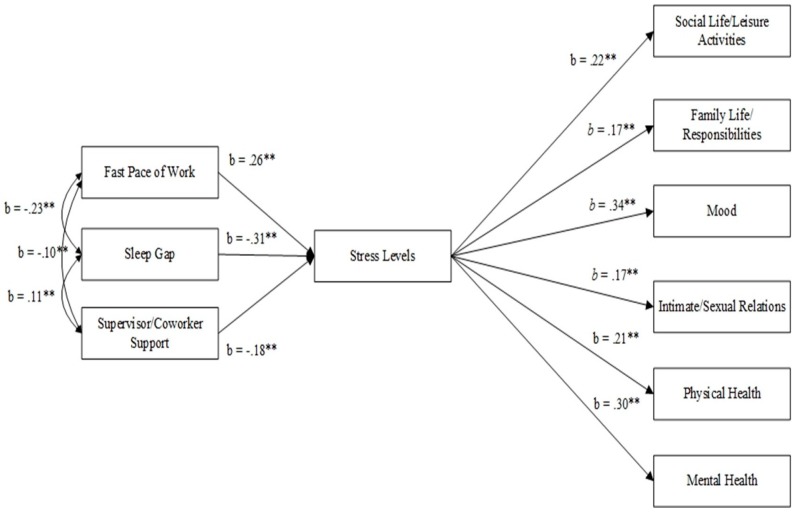
Structural equation model displaying the relationships between: pace of work; sleep gap; workplace social support; perceived job stress; and sleep’s impact on work-life balance. ******
*p* < 0.01.

**Table 1 ijerph-16-00984-t001:** Work organization.

Age. Driver Years of Experience, Work Organization Characteristics	*N* (%)
Age	
* 18–39*	69 (26.5)
* 40–49*	73 (28.1)
* 50 and older*	118 (45.4)
Years of Experience	
* Less than 10 years*	97 (37.3)
* 10–19 years*	79 (30.4)
* 20 or more years*	84 (32.3)
Days on road per month	
* 20 or less days*	40 (15.4)
* 21–25 days*	110 (42.3)
* 26 or more days*	110 (42.3)
Work hours per day	
* Less than 11 h*	77 (29.8)
* 11–13 h*	83 (32.1)
* 13 or more hours*	99 (38.3)
Daily schedule	
* Same each day*	45 (17.3)
* Different each day*	215 (82.7)
Hours of day	
* Same each day*	94 (36.2)
* Different each day*	166 (63.8)
Days of week	
* Same each week*	175 (67.6)
* Different each week*	84 (32.4)
Fast pace of work	
* Never or rarely*	83 (32.1)
* Sometimes*	56 (21.6)
* Often or always*	120 (46.4)
Coworker support	
* Never or rarely*	57 (30.0)
* Sometimes*	40 (21.1)
* Often or always*	93 (48.9)
Supervisor support	
* Never or rarely*	21 (8.4)
* Sometimes*	38 (15.3)
* Often or always*	189 (76.2)

**Table 2 ijerph-16-00984-t002:** Perceived stress and sleep outcomes.

Stress and Sleep Outcomes	Mean (SD)	Range	*N* (%)
Perceived stress			
* None–mild*			97 (37.3)
* Moderate*			104 (40.0)
* High or chronic*			59 (22.7)
Sleep duration in hours (Workdays)	6.95 (1.62)	3.0–13.0	
Sleep duration in hours (Non-workdays)	8.27 (2.12)	3.5–16.0	
Sleep duration needed for ‘highest function’	6.75 (1.53)	1.0–13.0	
Sleep quality (Workdays)—Frequency of ‘good night’s sleep’			
* Never or rarely*			98 (38.2)
* Almost or every night*			159 (61.8)
Sleep quality (Non-workdays)—Frequency of ‘good night’s sleep’			
* Never or rarely*			39 (16.7)
* Almost or every night*			194 (83.3)

**Table 3 ijerph-16-00984-t003:** Sleep’s impact on drivers.

Impact Outcomes	*N* (%)	Mean (SD)	Range
Impact on work			
* No impact*	48 (19.0)		
* Some impact*	111 (43.9)		
* Major impact*	94 (37.2)		
Impact on social and leisure activities			
* No impact*	94 (41.4)		
* Some impact*	78 (34.4)		
* Major impact*	55 (24.2)		
Impact on family and home responsibilities			
* No impact*	99 (40.9)		
* Some impact*	87 (36.0)		
* Major impact*	56 (23.1)		
Impact on mood			
* No impact*	46 (18.0)		
* Some impact*	106 (41.6)		
* Major impact*	103 (40.4)		
Impact on intimate and sexual relations			
* No impact*	122 (51.9)		
* Some impact*	66 (28.1)		
* Major impact*	47 (20.0)		
Impact on physical health			
* No impact*	93 (37.1)		
* Some impact*	87 (34.7)		
* Major impact*	71 (28.3)		
Impact on mental health			
* No impact*	94 (37.8)		
* Some impact*	87 (34.9)		
* Major impact*	68 (27.3)		
Work-Life Conflict		6.43 (4.30)	0–14
* None to minor impact (0–3)*	64 (30.5)		
* Mild impact (4–6)*	44 (21.0)		
* High impact (7–9)*	48 (22.9)		
* Major impact (10 or greater)*	54 (25.7)		

**Table 4 ijerph-16-00984-t004:** Associations between work organization, perceived stress and sleep’s impact on work-life balance (controlled for age and length of tenure).

Predictor Variables	Wald X^2^	OR	95% CI
25 or less days on road/month (reference: 26 or more)	0.97	0.76	0.44, 1.31
11 or less work hours per day (reference: more than 11)	0.46	0.83	0.47, 1.44
Same daily schedule (reference: different)	1.49	0.63	0.30, 1.32
Same hours per day (reference: different)	0.38	0.84	0.47, 1.48
Same days per week (reference: different)	0.02	1.04	0.59, 1.83
Frequency of fast pace of work			
* Never or rarely*	2.18	0.62	0.31, 1.18
* Sometimes*	0.02	1.05	0.53, 2.10
* Often or always (reference)*	-	-	-
Supervisor support			
* Often or always*	0.22	1.32	0.43, 3.99
* Sometimes*	1.21	1.52	0.72, 3.23
* Never or rarely (reference)*	-	-	-
Perceived Stress			
* None or mild (reference)*	4.18	0.45 *	0.21, 0.97
* Moderate*	4.72	0.45 *	0.22, 0.92
* High or chronic stress*	-	-	-

X = 34.39; *p* < 0.001; Cox and Snell R^2^ = 0.15; Nagelkerke R^2^ = 0.16; * *p* < 0.05.

**Table 5 ijerph-16-00984-t005:** Associations between Work Organization, Sleep, Work-Life Balance and Perceived Stress (controlled for age and length of tenure).

Predictor Variables	Wald *X*^2^	OR	95% CI
25 or less days on road/month (reference: 26 or more)	0.01	1.03	0.52, 2.04
11 or less work hours per day (reference: more than 11)	0.60	0.77	0.39, 1.50
Same daily schedule (reference: different)	0.14	1.18	0.51, 2.73
Same hours per day (reference: different)	1.22	1.48	0.74, 2.94
Same days per week (reference: different)	2.01	0.62	0.32, 1.20
Frequency of fast pace of work			
* Never or rarely*	18.56	0.18 ***	0.08, 0.39
* Sometimes*	11.59	0.24 ***	0.11, 0.55
* Often or always (reference)*	-	-	-
Supervisor support			
* Often or always*	0.02	0.91	0.23, 3.65
* Sometimes*	1.59	1.73	0.74, 4.07
* Never or rarely (reference)*	-	-	-
Sleep Duration (workdays)	14.62	0.60 ***	0.47, 0.78
Sleep Duration (non-workdays)	7.00	1.27 **	1.06, 1.52
Sleep Quality (workdays)			
* Almost or every night good sleep*	0.01	1.03	0.46, 2.30
* Never or rarely good sleep (reference)*	-	-	-
Sleep Quality (non-workdays)			
* Almost or every night good sleep*	4.71	0.32 *	0.11, 0.90
* Never or rarely good sleep (reference)*	-	-	-
Work-Life Balance (influenced by work/sleep)			
* None to minor impact*	1.06	0.64	0.27, 1.50
* Mild impact*	0.52	0.71	0.28, 1.81
* High impact*	0.01	1.04	0.44, 2.46
* Very high impact*	-	-	-

X = 65.01; *p* < 0.001; Cox and Snell R^2^ = 0.31; Nagelkerke R^2^ = 0.35; * *p* < 0.05; ** *p* < 0.01; *** *p* < 0.001.

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
