# Peer review of "Work-Life Conflict among U.S. Long-Haul Truck Drivers: Influences of Work Organization, Perceived Job Stress, Sleep, and Organizational Support"

_ijerph, 2019, doi:10.3390/ijerph16060984_

Round 1

Reviewer 1 Report

Dear authors of the manuscript "Work-life conflict among U.S. long-haul truck drivers: Influences of work organization, perceived job stress, sleep, and organizational support",

thank you for revising your manuscript in consideration of previous comments.

I read your manuscript again and still think that you provide a valuable contribution to the state-of-the-art in work-life research. Even though very selective and specific, your sample of U.S. long-haul truck drivers is not only a relevant group itself but has also working conditions that are transferable to other workers, especially in the course of the increasing importance of flexibility in time and place due to the digitalisation of work. This brings me to my only suggestion for further revision: Maybe your discussion-section could discuss the future relevance of your results or whether the results are transferable to other groups of workers that are similar to U.S. long-haul truck drivers because they work from home over long periods of time and have similar working conditions.

I wish you much success in publishing this research!

Author Response

We appreciate this reviewer’s continued support of our work. We have added in a couple of more sentences in the final conclusions section to connect with other similar groups of workers in terms of work conditions and the importance of future studies with other occupations.

Reviewer 2 Report

This paper focused on health risks and consequences in long haul truck drivers (LHTDs), particularly investigating  work, sleep, perceived stress, and their subsequent impacts on the work-life conflict of a sample of LHTDs.

Authors observe that perceived stress contributed to the relationship of poor sleep to worse work life conflict.

Authors use work and stress measures to see if they affect the outcome assessed as “impact of sleep on work-life conflict” (which were a specific set of questions on how perceived sleep problems influenced a variety of work and family outcomes). Wouldn’t it be simpler and more direct to use the replies to the specific sleep questions and see if they relate/predict the work/family outcomes? (they do this for the direct relationship of sleep to stress [Table 5]).

“Sleep gap” is an interesting outcome. Did authors create this, or has it been used in the literature elsewhere?

Why did authors decide to conduct the survey in n=262? Were there any power analyses/sample size estimates conducted to determine this number, or was this determined by some other factor?

Some more information on the Truck Sleep Disorders Survey (TSLDS) would be useful; has it been validated? Used by other groups?

LHTDs are asked in the questionnaire about their number of days worked, number of hours worked, and “irregularity” of shifts, but is there information about the shift types, i.e. whether the working was done during standard hours (9-5) or irregular/nonstandard hours, for example overnight, early mornings, etc. This would get at the shift work schedules mentioned in the Introduction.

For the stress assessment ("none, mild, moderate, high, 188 and extreme or chronic”) what is the timescale over which this is assessed?

Similar, for the sleep assessment, (number of hours slept over 24 hrs on work and nonwork days), how exactly is the question framed? For instance, are they responding over the past week, month? Or in general? While it is good to assess 24-hour sleep, it would be good to also have separate assessments for the main sleep episode (could occur during the daytime for night workers) and other naps separately.

Authors note an increase of > 1 hour in self-reported sleep duration on non-work days compared to work days. It would also be good to have information on the timing of sleep episodes, especially since “social jetlag” (difference in timing of midpoint of sleep between work and non-work days) has been shown to associate with several adverse physiological and psychological outcomes. This potential variability in the timing of sleep (presumably later on non-work vs work days) could theoretically relate to the surprising finding reported here that longer sleep on non-work days was related to higher odds of perceived stress.

Rationale/justification for using structural equation modeling on cross-sectional data to explore for potential mediating influences seems reasonable/justified (I am not a statistical expert)

Support from supervisors and coworkers is assessed, but what about family/friends support?

Use of self-reported sleep and lack of objective sleep assessment (e.g. via accelerometry) might also be considered a limitation.

Author Response

We thank the reviewer for their comments and suggestions for improvement. Please see our responses to the reviewer below.

1. In terms of the question regarding the sleep questions and a direct relationship with the work/family outcomes, we had tested the model with the sleep variables included. It was not statistically significant, so based on that analysis, we removed the sleep variables and fit a statistically significant model. We have now added in more language in the Statistical analyses section to describe the rationale. This is where the relationships between sleep, work organization/scheduling, occupational stress, and impact of sleep on work-life conflict provides an interesting relationship – as detailed in our model hypothesis and confirmed in our SEM model.

2. The sleep gap has been operationalized and used in the National Sleep Foundation “Sleep in America” polls. This helps researchers measure the amounts of sleep that Americans are getting in relation to what they would like to get to be fully functional on a daily basis. We have now added a sentence in and have cited the National Sleep Foundation.

3. Prior to the study and data collection, power analyses were utilized to detail the sample size of the study. We have now added in a sentence in the methods to confirm that for the reader.

4. The extended details of the study design have been provided in our previous papers using this data. We reference these papers in the methods section. The TSLDS was validated prior to it being used. It was tested on a smaller sample of drivers at the truck stop where data collection took place and analyzed by other public health researchers. Upon the analyses and testing with drivers, we were able to determine construct validity, identify and add missing items, clarify scale distributions, conduct item correlations, and determine the reliability. The TSLDS has not been utilized by other groups, as of yet.

5. In our study, we did not ask drivers about their shifts in terms of when, specifically they occur. That is a limitation of our study, and we seek to measure that in future studies.

6. This is a cross-sectional study and the timescale regarding the perceived stress of drivers is just the one “snapshot” in time. We just simply asked drivers, “How would you perceive your stress level regarding your job?”

7. The questions regarding sleep were asked in this format: “over the past two weeks….?” We added in a sentence in the study measures section to clarify that. We did not specify between naps, etc. – we just wanted to know about sleep in total over the 24 hour time period.

8. We did not measure for “social jetlag” in this particular study. However, that is a great idea for future studies, and we will seek to measure that moving forward. We agree that there is a strong likelihood that sleep on non-workdays adds to stress for drivers as they are trying to catch up on missed sleep/rest during the workweek. It also takes away from their out-of-work activities and responsibilities.

9. In this study, we did not ask about friends/family support. However, that is a great recommendation for future studies.

10. Yes, the use of subjective sleep measures is a limitation of this study.

This manuscript is a resubmission of an earlier submission. The following is a list of the peer review reports and author responses from that submission.

Round 1

Reviewer 1 Report

This paper presents an interesting setting for a study on stress, work-life balance and wellbeing. 

However, I found  that the theoretical framework and theoretical contribution are not clear. In this paper, work-life balance seems to refer to work-life conflict, which is one aspect of work-life balance. There is already significant empirical evidence that associates work-life conflict with stress, negative health outcomes and general well-being. Antecedents to work-life conflict (e.g., work demands) are also well researched. There is insufficient information in the paper to understand the basis for the research question, its relevance and expected contribution to theory and practice.

I also found that hypotheses are not established and thus, without knowing what it is being tested, it is not possible to evaluate the soundness of the method.

I would suggest to review the work-life balance literature, in particular, work-life conflict, and identify a relevant research question and expected theoretical contribution.

Author Response

We have now added in further literature in the introduction to support our theoretical framework, which includes elements of work organization, work-life conflict, and organizational support/safety climate. The combination of these three elements have profound implications for LHTD sleep and health outcomes, as well as public safety on roadways; closely linked to it, we hypothesized that it would have significant impacts on how drivers perceive their job stress and subsequent work-life balance or conflict.

With this, we appreciated the suggestion to focus primarily on work-life conflict, which is most problematic among LHTDs; furthermore, with the data set that we have, these were the only aspects for us to consider. We do not have any data related to specific family impacts, etc. In response, we have added another paragraph in the introduction to detail the mediating/moderating role of organizational support/safety climate in LHTD. This helps to further support our 2 hypotheses and the post-hoc test of our path model. With this, we also added that LHTDs are considered “lone workers”, which makes this context really unique in studying.

This study, by making use of elements of the three theoretical perspectives, can provide more effective and specific recommendations for policy and organizational support to promote and protect LHTD health and safety.

Reviewer 2 Report

This paper sought to explore relationships between work, sleep, perceived stress, and their subsequent impacts on the work-life balance of U.S. long-haul truck drivers. I think the topic of the paper is interesting and important. Structural equation modeling can not only explore for potential mediating influences and can estimate direct effect. However, this paper only considers mediating influences in structural equation modeling. In addition, each of the individual work-life balance indicators may be related to each other. I am confused why not consider constructing a second-order model.

Author Response

The researchers appreciate this comment and perspective. In response, with this study being exploratory, we were only focused on the work-life conflict indicators. In the logistic regression models (Tables 4 & 5), we did make use of the work-life variables in a composite variable; in the methods, we highlighted that the work-life variables were highly related to each other (Chronbach’s alpha = 0.90). Therefore, in the path analysis, we desired to “tease” those out as individual factors to see how each was impacted by the organizational climate, work organization, sleep, and stress variables. From the path models, we could detail how mental health, mood, and social life/leisure activities were most impacted by drivers’ perceived stress.

As stated, we appreciate the suggestion and will make use of this recommendation in future studies, but we did not deem it appropriate or the focus of the current study. 

Reviewer 3 Report

Dear authors of the manuscript “Work-life balance among U.S. long-haul truck drivers: Influences of work organization, perceived job stress, and sleep”,

I am convinced that you research an interesting and relevant topic that contributes to previous literature. The focus as well as the goal of this study are relevant and the methods and results adequately described. I have some advice for revisions that I recommend to consider when revising the manuscript:   

1. Introduction: Overall, the introduction is well written. However, I think it would be interesting to learn a little more about the group of U.S. long-haul truck drivers in this part of the paper. I would be interested in what is – according to this or other research – the “typical” LHTD. To which social or ethnic group does he belong, how old is he or what family status does he have. Moreover, I would recommend the authors to extend the introduction in respect of what theoretical assumption the study is based on. Do you use and test a specific theoretical model? This has not become completely clear to me. Finally, the approach to work-life balance to some degree always includes the idea that resources from the job or the private and family life are able to reduce or even buffer the negative effects of job stressors. Regarding the very selective and specific group of LHTDs, I think it would be interesting to discuss some ideas whether LHTDs differ in the resources they have. Do they generally have fewer resources because they are always away from home? Or do they have specific resources such as high social support from other LHTDs? This is also something that could be discussed in the discussion section.

2. Methods: It seems to me that your analysis lacks some important control variables that may be of importance for both health and WLB. For example, different age groups differ in their experience of WLB as well as in their sleep outcomes. In addition, it would be interesting to consider (if you have this in your data) how long the LHTD is doing this job. Most importantly, I would like to point to the fact that if you talk about WLB, this always includes work as well as private or family life. However, you focus on work characteristics but I would recommend to consider private  or family conditions such as the partnership status, having children or other care responsibilities that are significantly important for individual work life balance.

I hope that my comments and recommendations help you revising the manuscript. I wish you much success with this research!

Author Response

1.      We appreciate all of these wonderful and thoughtful recommendations. We have now included additional information about the “typical” LHTD in the third paragraph of the Introduction. In our introduction, we have also now expanded upon our theoretical framework, which includes elements of work organization, work-life balance/conflict, and safety climate (which focuses on support from the organization and policy aspects of work). In terms of the safety climate, we hypothesized in our path model that stress would play a mediating role between safety climate and how drivers perceived their work-life conflict (which was confirmed). In addition, these aspects were affected by poor sleep outcomes and the pace of work experienced by LHTDs due to work organization and scheduling practices. Unfortunately, we do not have data to further explore the elements of resources offered from the job or family life as the ability to buffer stress. However, we have now added in the introduction and discussion about how LHTDs lack support resource and healthcare resources often found in work settings (due to being away from home for extended periods of time), due to them being characterized as ‘lone workers’, who often perform their job duties with limited interaction with coworkers.

2.      In previous studies, we have examined the role of age groups in relation to health outcomes (obesity, sleep, etc.). In preliminary analyses for this study, we did account for age. However, in this study focused on work-life influence, age had no effect on the models and the models were actually stronger without age included. Therefore, in our final models, we excluded age. In addition, age is an interesting factor in this population; those that come in at an early age and “cannot handle” the poor working conditions, typically leave the profession, whereas, those that remain in the profession typically have fewer complaints. In terms of the other variables that were suggested, unfortunately, we did not collect this information in the present study. Our work-life variables do ask the broad questions related to work/sleep’s impact on family and time outside of work; however, we did not collect any data in relation to partnership status, children, or specific out of work care responsibilities. This is a great suggestion and we will seek to do so in future studies of LHTDs!

Round 2

Reviewer 1 Report

Based on my initial comments, the revised paper still requires significant work in order to demonstrate a clear theoretical contribution and rigour of method. 

Reviewer 2 Report

An interesting contribution. The conclusions drawn by the authors seem to me plausible enough to believe. I think the paper has some value, so it should not be thrashed.